Estimating the impact of climate change on the potential distribution of Indo-Pacific humpback dolphins with species distribution model

Fu Jinbo 1
Zhao Linlin 2
Liu Changdong changdong@ouc.edu.cn 1
Sun Bin 1
1 Department of Fisheries, Ocean University of China , Qingdao , Shandong , China
2 First Institute of Oceanography, Ministry of Natural Resources , Qingdao , Shandong , China
Nazareno Alison
Electronic publication date: 2021 Aug 17
Publication date: 2021
Volume: 9
Electronic Location ID: e12001
Received 2021 May 13; Accepted 2021 Jul 27
Copyright: ©2021 Fu et al.
Copyright year: 2021
Copyright holder: Fu et al.
License: This is an open access article distributed under the terms of the Creative Commons Attribution License, which permits unrestricted use, distribution, reproduction and adaptation in any medium and for any purpose provided that it is properly attributed. For attribution, the original author(s), title, publication source (PeerJ) and either DOI or URL of the article must be cited.
License URL: https://creativecommons.org/licenses/by/4.0/

Keywords: Sousa chinensis, Marine fauna, Greenhouse gas emissions, Climate impact, Future projections

Funding: The authors received no funding for this work.

==============================
As IUCN critically vulnerable species,the Indo-Pacific humpback dolphins (Sousa chinensis) have attracted great public attention in recent years. The threats of human disturbance and environmental pollution to this population have been documented extensively. However, research on the sensitivity of this species to climate change is lacking. To understand the effect of climate change on the potential distribution of Sousa chinensis, we developed a weighted ensemble model based on 82 occurrence records and six predictor variables (e.g., ocean depth, distance to shore, mean temperature, salinity, ice thickness, and current velocity). According to the true skill statistic (TSS) and the area under the receiver operating characteristic curve (AUC), our ensemble model presented higher prediction precision than most of the single-algorithm models. It also indicated that ocean depth and distance to shore were the most important predictors in shaping the distribution patterns. The projections for the 2050s and 2100s from our ensemble model indicated a severe adverse impact of climate change on the Sousa chinensis habitat. Over 75% and 80% of the suitable habitat in the present day will be lost in all representative concentration pathway emission scenarios (RCPS) in the 2050s and 2100s, respectively. With the increased numbers of records of stranding and deaths of Sousa chinensis in recent years, strict management regulations and conservation plans are urgent to safeguard the current suitable habitats. Due to habitat contraction and poleward shift in the future, adaptive management strategies, including designing new reserves and adjusting the location and range of reserves according to the geographical distribution of Sousa chinensis, should be formulated to minimize the impacts of climate change on this species.

Introduction

As the most concerning environmental issue, global climate change has caused significant changes in marine environmental conditions over the past decades (Belkin, 2009; Cheung, Watson & Pauly, 2013; Wu, 2020). For instance, the assessment that was made for the coastal China seas over the 21st century shows that the East China Sea (ECS) will be simultaneously exposed to enhanced warming, deoxygenation, acidification, and decreasing net primary productivity (NPP) as a consequence of increasing greenhouse gas emissions (Tan et al., 2020). A species lives on a certain environmental niche space, so the change of the dependent environment conditions may change the distribution of this species (Bellard et al., 2012; Faleiro, Nemesio & Loyola, 2018). The distributions of marine mammals have been impacted significantly by the environmental change over the past decades (Nøttestad et al., 2015; Chen et al., 2020). According to these facts, understanding how future climate change will influence species distributions is vital for the better protection of species.

Species distribution models (SDMs) build species-environment relationships that are typically based on species location data (e.g., abundance and occurrence) and environmental variables that are thought to influence species distributions and can provide a useful framework for identifying and evaluating the habitat suitability for a given species (Guisan & Thuiller, 2005). Currently, SDMs are applied broadly in the life and environmental science fields (Cheung et al., 2009; Robinson et al., 2011). For example, multiple types of SDM are available to predict the impacts of climate change on species distributions (Zhang et al., 2019), to assess how habitat loss restricts large-scale species distribution (Vasconcelos & Doro, 2016)), to understand biological invasions (Zhang et al., 2020a) and to site aquaculture farms (Dong et al., 2020). Accordingly, the use of SDMs in conservation biology and biodiversity assessments is ever-increasing (Araujo et al., 2019; Zhang et al., 2020b).

Indo-Pacific humpback dolphins (Sousa chinensis), also known as “mermaids” and “water pandas”, belong to the porpoise family of cetaceans (Jefferson & Karczmarski, 2001). Due to their preferred inshore and estuarine habitats, Sousa chinensis are typically found in the shallow, coastal waters of the Indian and western Pacific oceans (Jefferson & Rosenbaum, 2014; Jefferson & Smith, 2016; Parra & Jefferson 2018). These areas, which have intensive commercial fisheries, are usually rapidly developing and are easily polluted by industrial production and daily activities of residents (Chen et al., 2008; Wu & Chen, 2014; Karczmarski et al., 2016; Guo et al., 2020; Li, 2020); the corresponding consequences of habitat degradation may lead to population declines of Sousa chinensis or even put this species at risk of extinction. In recent years, the numbers of records on strandings or deaths of Sousa chinensis have increased in China (Chen et al., 2008; Guo et al., 2020). This species has already been classified as “vulnerable” by the International Union for Conservation of Nature (Jefferson et al., 2017). Consequently, formulating a conservation plan for Sousa chinensis is urgent under current and future environmental scenarios. Predicting the geographical distribution for the present and future is a prerequisite for plan formulation (Schickele et al., 2020).

In this study, we developed ten individual SDMs. We then built a weighted average ensemble model, which has not been used to identify the potential distributions of Sousa chinensis under present-day and future climate scenarios. Our ensemble model was expected to present a better performance in predictive accuracy and uncertainty reduction than each individual model. We hypothesize the suitable habitats of Sousa chinensis will contract and shift poleward because of climate change in the future. Our ensemble model can help us to (1) determine the important environmental variables that affect Sousa chinensis distributions, (2) map the environmental suitability for Sousa chinensis under present-day and future climate scenarios, and (3) assess the impacts of climate change on Sousa chinensis habitat distributions. Our study can provide important implications for formulating adaptive management strategies, including designing new reserves and adjusting the location and range of reserves according to the geographical distribution of Sousa chinensis under current and future scenarios and an important reference to solve marine conservation planning problems. It also provides guidance for research on the potential distributions of other protected species under future climate change scenarios.

Materials and Methods

Study area and Sousa chinensis data collection

Indo-Pacific humpback dolphins (Sousa chinensis) are mainly distributed in the Western Pacific and the Indian Ocean, so our research is located in these areas (i.e., 50° E to 180° E, 50° S to 50° N; Fig. 1). Georeferenced species data (presence) were obtained from the online database: Global Biodiversity Information Facility (GBIF, https://www.gbif.org) and Ocean Biogeographic Information System (OBIS, https://obis.org). The cluster samples in a 5 × 5 arc-minute grid consistent with the spatial resolution of environmental data are removed; only one record per grid unit is used to avoid over-representation of environmental conditions (sampling bias) in densely sampled areas. A total of 124 incidents were retrieved, 82 of which were within our study area.

Figure 1 Binary output of habitat suitability and predicted potential distribution under current climate conditions of Sousa chinensis.

(A) Binary output of habitat suitability under current climate conditions. (B) Predicted current potential distribution. Green colors indicate suitable areas, and gray colors represent unsuitable ranges on the left; the color gradient indicates variations in habitat suitability on the right (green = highest and pink = lowest); the purple dots show the occurrence points that were used to develop the species distribution model.

Environmental variables and future projections

The raster data of environmental variable projections in this study were retrieved from the Bio-ORACLE v2.1 dataset (http://www.bio-oracle.org) (Assis et al., 2018) and Global Marine Environment Datasets (http://gmed.auckland.ac.nz) (Basher, Bowden & Costello, 2014). The mean chlorophyll, velocity, salinity, temperature, dissolved oxygen content, ice thickness, and pH data were obtained from Bio-ORACLE. The distance to shore and mean ocean depth data were obtained from GMED. In addition, the annual ranges of chlorophyll, flow rate, salinity, temperature, dissolved oxygen, and ice thickness were also obtained from Bio-ORACLE. There were a total of 15 environmental variables with a spatial resolution of 5 × 5 arc-minutes (i.e., 9.2*9.2 km at the equator). Pearson correlation analysis was conducted for these 15 environmental variables to reduce the influence of collinearity on the precision of model predictions. By comprehensive consideration of the availability of current and future environmental data, six low-correlation (pairwise Pearson’s correlation coefficients were less than —0.7—) (Dormann et al., 2013) environmental variables, including mean current velocity, mean salinity, mean temperature, mean ice thickness, mean ocean depth and distance to shore, were finally selected for the modeling analysis (Fig. S1).

Meanwhile, the projections of the first four environmental variables for the future (i.e., 2040–2050 (the 2050s) and 2090–2100 (the 2100s)) under four representative concentration pathway emission scenarios (RCPS) were also retrieved from the Bio ORACLE v2.1 dataset. RCPs (i.e., RCP26, RCP45, RCP65, and RCP85) are new climate change scenarios on radiation forcing at the end of the 20th century that were published in the fifth assessment report of the Intergovernmental Panel on Climate Change (IPCC). RCP26 indicates an optimistic emission level resulting in low greenhouse gas concentration; RCP45 and RCP60 represent the moderate emission level; and RCP85 indicates an pessimistic emission level leading to the highest greenhouse gas concentration (Moss et al., 2010). We assumed that distance to shore and ocean depth remain constant in the future. Future temperature, salinity, ice thickness, and current velocity) in Bio-ORACLE were predicted based on the mean simulation results of three Atmosphere-Ocean General Circulation Models (AOGCMs), i.e., Community Climate System Model (CCSM4, National Center for Atmospheric Research), Hadley Center Global Environment Model, version 2 (HadGEM2-ES, Met Office Hadley Centre) and Model for Interdisciplinary Research on Climate, version 5 (MIROC5, Atmosphere and Ocean Research Institute, National Institute for Environmental Studies, and Japan Agency for Marine-Earth Science and Technology) from the Coupled Model Intercomparison Project 5 (CMIP 5) (Assis et al., 2018; Sharma et al., 2021). We believed the mean results of three different AOGCMs can reduce the uncertainties effectively. The changes in the four predictor variables in the future (i.e., the 2050s and the 2100s) under different scenarios are shown in Table 1.

Table 1 Current environmental conditions and the averages and ranges of climatic changes for the future (i.e., the 2050s and 2100s) under different scenarios in the study area.

Environment
variable	Current value	Changes in 2050s	Changes in 2100s	
		RCP26	RCP45	RCP60	RCP85	RCP26	RCP45	RCP60	RCP85	
T(° C)	22.72	0.72
(0.19,1.89)	0.96
(0.06,2.35)	0.77
(0.81,1.76)	1.10
(0.50,2.30)	0.63
(0.80,1.87)	1.21
(0.24,2.67)	1.68
(0.47,3.42)	2.87
(1.56,5.53)	
Sal(PSS)	34.51	−0.061
(−0.12,0.09)	−0.07
(−0.70,0.45)	−0.07
(−0.91,0.23)	−0.07
(−0.88,0.33)	−0.09
(0.88,1.15)	−0.13
(−1.03,0.40)	−0.16
(−1.64,0.42)	−0.26
(−1.97,0.53)	
CV(m/s)	0.10	0.00
(−0.06,0.09)	0.24
(−0.84,1.68)	0.25
(−0.84,1.66)	0.00
(−0.12,0.09)	0.24
(−0.85,1.68)	0.13
(−0.84,1.67)	0.13
(−0.84,1.67)	0.23
(−0.84,1.68)	
Ice(m)	0.00	0.00
(−0.10,0.00)	0.00
(−0.12,0.00)	0.00
(−0.10,0.00)	0.00
(−0.13,0.00)	0.00
(−0.12,0.00)	0.00
(−0.17,0.00)	0.00
(−0.17,0.00)	0.00
(−0.17,0.00)	
Notes.

T temperature

Sal salinity

CV current velocity

Ice ice thickness

The values in parentheses indicate the minimum and maximum change of climate

Modeling procedures

We conducted the model analysis on the R platform (R Development Core Team, 2020) based on the “biomod2″package, and ten SDMs were available in this package (Thuiller, Georges & Engler, 2020). The ten models include the generalized linear model (GLM) (McCullagh & Nelder, 1989), generalized additive model (GAM) (Hastie & Tibshirani, 1990), classification tree analysis (CTA) (Breiman et al., 1984), generalized enhanced regression model (GBM) (Ridgeway, 1999), artificial neural network (ANN) (Lek & Guegan, 1999), surface range envelope (SRE) (Breiman, 2001a), flexible discriminant analysis (FDA) (Hastie, Tibshirani & Buja, 1994), multiple adaptive regression splines (MARS) (Friedman, 1991), random forest (RF) (Breiman, 2001b), and maximum entropy model (Maxent) (Phillips, Anderson & Schapire, 2006).

Due to the lack of true absence records, we simulated 5000 pseudo-absence points randomly in contrasting environmental conditions with the true presence points (Guisan, Thuiller & Zimmermann, 2017; Thuiller, Georges & Engler, 2020). A fivefold cross-validation technique with 10 repetitions was used to assess the model prediction accuracy (Guisan, Thuiller & Zimmermann, 2017; Thuiller, Georges & Engler, 2020). Based on this approach, 80% of the dataset was randomly selected for calibration and testing of the models, and 20% was withheld for evaluation of the model predictions. Two indicators were used to evaluate the predictive capability of each model: the true skill statistic (TSS) (Allouche, Tsoar & Kadmon, 2006) and the area under the receiver operating characteristic curve (AUC) (Swets, 1988). To ensure sufficient prediction accuracy, the models with mean TSS value above 0.80 and mean AUC value above 0.85 were reserved for further analyses (Zhang et al., 2019).

According to the cutoff values of TSS and AUC, the retained individual SDMs were used to build a weighted average ensemble model predicting the Sousa chinensis distributions under present and future climate conditions based on the “biomod2” package (Zurell et al., 2020). For a better interpretation of model outcomes, continuous habitat suitability projections were converted into binary maps (e.g., suitable/unsuitable) by using an automatically generated threshold that maximized the TSS value of the ensemble model (Liu, White & Newell, 2013; Guisan, Thuiller & Zimmermann, 2017; Zhang et al., 2020b).

The relative importance of each environmental variable in predicting the Sousa chinensis distributions was determined by a randomized approach. This approach computes the Pearson correlations among predictions using all predictor variables and predictions in which the predictor variable being evaluated was randomly permutated (Guisan, Thuiller & Zimmermann, 2017; Thuiller, Georges & Engler, 2020). Low correlations between the standard predictions and those using the permuted variable indicate the high importance of a predictor variable (Zhang et al., 2019). A response curve, which describes the variations in species occurrence probability along the gradient of each important predictor variable, was plotted.

Results

Model performances and predictive accuracy of SDMs

The different AUC and TSS values indicated the different predictive performances among all 10 modeling algorithms. All the models except SRE, MAXENT and FDA exhibited good predictive capacity and were selected to construct the ensemble model (Figs. S2 and S3). The AUC and TSS values of all the individual models except GBM and RF were lower than those of the ensemble model (AUC: 0.993, TSS: 0.963), which demonstrated the superior predictive performance of the ensemble model.

Response curve and variable importance

The six predictor variables made different contributions to the Sousa chinensis distributions. Among the six predictor variables, depth (0.435 ± 0.029) and distance to shore (0.473 ± 0.031) were the two most important variables for the model predictions. The contributions of temperature (0.234 ±  0.018), salinity (0.135 ± 0.013), and current velocity (0.080 ± 0.011) were moderate, while ice thickness (0.003 ±  0.0007) was considered to be nearly irrelevant (Fig. S4). The response curves of Sousa chinensis to the three most important variables from the ten models (except SRE) are shown in Fig. S5. The response curves indicated that the environmental requirements of Sousa chinensis in the different models were generally similar.

Potential distributions under present and future climate scenarios

Our prediction of suitable habitat for Sousa chinensis under present climate conditions is shown in Fig. 1. All of the occurrence records were within the predicted suitable range. The predictions show that a large part of the coastal areas of the Southeast Asian countries and northern Australia are suitable habitats for Sousa chinensis. Some of the occurrence records were located in the coastal areas of the Indian Peninsula.

As the model results show, the suitable area for Sousa chinensis will decrease under all four assumed future climate change scenarios. Future habitat projections under different RCP scenarios show different distribution patterns and consistently suitable range contraction for Sousa chinensis (Table 2). The model projections indicate that the contraction of the suitable range of this species could be from 75.626% (under the RCP2.6 scenario in the 2050s) to 95.815% (under the RCP8.5 scenario in the 2100s). Future predictions for the 2100s show that environmental conditions suitable for Sousa chinensis will shift northward to the East China Sea and south coast of Japan. The equatorial sea area and coastal area of northern Australia are predicted to be less suitable for this species (Fig. 2).

Table 2 Variation of distribution range (%) of Sousa chinensis under future climate scenarios.

	2050s	2100s	
	RCP26	RCP45	RCP60	RCP85	RCP26	RCP45	RCP60	RCP85	
PercLoss	75.626	84.197	81.133	81.493	80.566	86.451	90.72	95.815	
PercGain	3.995	3.507	3.595	3.86	3.595	3.562	3.653	5.083	
PercStable	24.374	15.803	18.867	18.507	19.434	13.549	9.28	4.185	
SpeciesRangeChange	−71.631	−80.69	−77.538	−77.633	−76.971	−82.889	−87.067	−90.732	
Notes.

RCP representative concentration pathway

PercLoss percentage of loss

PercGain percentage of gain

PerStable percentage of stability

Species range changes were calculated as (suitable range under future climate scenarios—present-day suitable range)/present-day suitable range.

Figure 2 Range shifts in habitat suitability of Sousa chinensis as projected by the ensemblespecies distribution model between current and future climate conditions.

(A) Under the RCP2.6 scenario in the 2050s, (B) under the RCP8.5 scenario in the 2050s, (C) under the RCP2.6 scenario in the 2100s, and (D) under the RCP8.5 scenario in the 2100s. Purple indicates areas that will become suitable in the future, dark yellow areas are projected to besuitable under both present-day and future climates, and black represents suitable areas that will become unsuitable in the future.

Discussion

Model performance

Utilizing georeferenced presence/pseudo-absence data and the corresponding environmental data, we developed an ensemble model for Sousa chinensis to predict the present and future potential distributions of this rare species. The results demonstrate that our ensemble model performed well in predicting the habitat suitability for Sousa chinensis under the present environmental conditions. The model predictions indicated that the potential distribution of Sousa chinensis will contract in the future under all the RCP scenarios and that the suitable habitat in the Indo-Pacific Mid-Seas will shift to higher latitudes.

Many mature models can be used to predict species distributions. The most commonly used method is to select the best model based on performance indicators such as TSS and AUC and then use the single best model to predict species distributions. Due to the higher accuracy and reliability compared to individual models, several published studies recommended using ensemble model to predict potential species distribution and habitat suitability (Araujo & New, 2007; Thuiller et al., 2009; Shabani, Kumar & Ahmadi, 2016). In this study, the weighted ensemble model performed better than most of the individual models but was not the best in predictive performance, and this finding was consistent with the previous study (Hao et al., 2020). The ensemble model is built based on the weighted average of individual models, so this model will present an advantage over individual models in reducing the uncertainties of model results.

Climate change and associated distribution shift

The predicted suitable habitats of Sousa chinensis include their known distribution range as expected (e.g., the coast of Malaysia). Suitable habitats were also found beyond where the species have been recorded, and this phenomenon can be caused by many factors, such as biotic interactions, dispersal limitation of species, niche size, and sampling bias (Pulliam, 2000; Goldsmit et al., 2018). Published studies have reached similar conclusions in predicting species distributions using SDM (Goldsmit et al., 2018; Zhang et al., 2020a). As shown in the binary output of habitat prediction, the main Sousa chinensis habitat in China is located in the Pearl River Estuary in Guangdong Province. The Pearl River Estuary is an intersection area of brackish and freshwater that results in fertile water quality and high primary productivity. The suitable temperatures and salinities, as well as the low pollution, high biodiversity, and unexploited natural shorelines, all make this area a favorite for Sousa chinensis.

According to the projected layer of future climate that was produced from 3 distinct AOGCMs provided by CMIP 5, we determined the changes in four available environmental variables. As shown in Table 1, temperatures will increase with different amplitudes under different RCPs. This tendency of global warming will severely affect Sousa chinensis distributions in terms of range size, e.g., it will probably lead to a reduction of more than four-fifths of its range in the 2100s. Meanwhile, the suitable Sousa chinensis habitat in the future will shift northward. In China, the suitable habitat on the southern coast will shift to the east Yellow Sea and even to the coastal areas of Bohai Bay. Tan et al. (2020) assessed the East China Sea (ECS). They found that climate change caused by increasing greenhouse gas emissions will induce considerable biological and ecological responses and cause the ECS to be among the ocean areas that are most vulnerable to future climate change. For example, the rising sea temperature and the change of dissolved oxygen content in the ECS affected the metabolic process of marine organisms and brought significant changes in the abundance and geographical distribution of marine life (Stenseth et al., 2002; Walther et al., 2002). On the other hand, the habitat in areas around Australia will shift southward in the future. The areas off the coast of Malaysia will no longer be suitable for Sousa chinensis. This trend toward higher latitudes is similar to that described in the formal research (e.g., Ruiz-Navarro, Gillingham & Britton, 2016; Zhang et al., 2019; Zhang et al., 2020c). Regardless of the dispersal scenario, our results highlight the high vulnerability of this critically vulnerable species to climate change.

Impact factors of Sousa chinensis distribution

Due to the intricate relationships among survival, growth and environmental conditions, many factors may affect the habitat distributions of Sousa chinensis. The basic niche that is suitable for the growth of Sousa chinensis, such as water temperature, water depth, and distance from shore, was considered in this study. The distribution of Sousa chinensis is negatively correlated with distance from shore and distance from the main estuary (Chen et al., 2020); hence, estuaries have been identified as their preferred habitat (Jefferson & Karczmarski, 2001; Chen et al., 2008; Jefferson & Smith, 2016). Environmental change induced by climate change may also affect the distributions of bait fishes and will indirectly affect Sousa chinensis distributions (Schickele et al., 2020).

Human activities have great impacts on Sousa chinensis habitats. The coastal areas of the China Sea, with many estuaries, bays, coral reefs and fisheries, are not only suitable habitats for Sousa chinensis but are also the most active areas for developing the maritime economy. Fishing behavior and boat travel have been determined to cause stranding deaths of Sousa chinensis (Guo et al., 2020). Sousa chinensis proved to be more acoustically active and prefer locations with lower noise levels (Caruso et al., 2020a; Caruso et al., 2020b). However, human activities often generate underwater noise, which interferes with information exchange with conspecifics and interaction with the surrounding environment and can even lead to behavioral disorders (Xu et al., 2020). Meanwhile, Sousa chinensis prefer waters near the natural coastline, while human activities such as sea reclamation would change the type of coastline and reduce the length of the natural coastline. Since the middle of the last century, the proportion of natural coastlines in China has continued to decline (Hou et al., 2016), which makes it more difficult for Sousa chinensis to find their preferred habitats and makes this sensitive species more vulnerable to extinction.

Conservation suggestions

Our results indicated that over 75% and 80% of the suitable habitat for Sousa chinensis in the present-day would be lost in all RCP scenarios in the 2050s and 2100s, respectively. The results may be inflated by the high over prediction of the models due to the few environmental variables used and not accounting for the biotic variables influencing Sousa chinensis distribution. Meanwhile, areas, where the species have never been observed in the current climate conditions and were lost in the future climate scenarios may also inflate the results. However, the change of marine environment induced by climate change will undoubtedly cause habitat reduction and poleward in the future. Therefore, adaptive management strategies are important for minimizing the impact of climate change on this vulnerable species.

Protected areas have been considered to be effective in situ strategy for conserving biodiversity and ecosystem services (Wang & Li, 2021). As a vulnerable species with great public concern, conservation attention has been given to Sousa chinensis. Seven natural reserves have been set up for this species (Indo-Pacific Humpback Dolphins Conservation Program (2017–2026)) in China. The adverse effects of climate change on the protected areas of many amphibian species and Chinese giant salamander (Andrias davidianus) have been elucidated (e.g., (D’Amen et al., 2011; Zhang et al., 2020b). The same situation will possibly occur in the protected areas for marine mammals such as Sousa chinensis. For instance, Hunt et al. (2020) used SDMs to predict the Sousa chinensis distribution in the marine reserve in Australia and evaluated that the effect in the established reserve will be reduced. Based on the present prediction of Sousa chinensis distribution in this study, we suggest establishing new reserves and limiting human and industrial activities in the important habitats in the short term. Based on the predicted distribution for future, we suggest formulating adaptive management strategies, including restoring the damaged habitats, adjusting the reserve range and location according to the change of distribution, and pre-establishing conservation areas in the long term. For example, the coastal areas of the East Yellow Sea in China should be protected to prepare for the possible poleward shift of Sousa chinensis distribution.

Conclusion

In this study, we developed a weighted average ensemble model based on 82 occurrence records and six predictor variables to predicted the potential distribution of Sousa chinensis under current and future climate scenarios. Our results indicated that over 75% and 80% of the suitable habitats in the present-day would be lost in all RCP scenarios in the 2050s and 2100s, respectively. The contraction and shift poleward of suitable habitats in the future imply that adaptive management strategies are important for minimizing the impact of climate change on Sousa chinensis. The results from our study can be used as references to formulate specific protection plans, such as designing new reserves and adjusting the current reserves.

Supplemental Information

Supplemental Information 1 Presence data worldwide

Click here for additional data file.

Supplemental Information 2 Presence data in study area

Click here for additional data file.

Supplemental Information 3 R script

Click here for additional data file.

Supplemental Information 4 The six environmental variables selected for building species distribution models

(A) mean temperature, (B) mean ocean depth, (C) distance to shore, (D) mean current velocity, (E) mean ice thickness, and (F) mean salinity.

Click here for additional data file.

Supplemental Information 5 The area under the receiver operating characteristic curves (AUC) of 10 modeling algorithms and the ensemble model that were used to estimate the habitat suitability of Sousa chinensis

Dashed line represents the threshold for AUC (0.85) to build the ensemble model. Dotted line represents the AUC value of the ensemble model.

Click here for additional data file.

Supplemental Information 6 The true skill statistics (TSS) of 10 modeling algorithms and the ensemble model that were used to estimate thehabitat suitability of Sousa chinensis

Dashed line represents the threshold for TSS (0.8) to build the ensemble model. Dotted line represents the AUC value of the ensemble model.

Click here for additional data file.

Supplemental Information 7 Variable importance of the six predictor variables from the 10 species distribution models for Sousa chinensis

T: temperature, Depth: ocean depth, Dshore: distance to shore, CV: current velocity, Ice: ice thickness and Sal: salinity. Data are expressed as means ±  standard error.

Click here for additional data file.

Supplemental Information 8 Sousa chinensis response curves for the nine spatial distribution modeling techniques against depth, temperature, and distance to shore

(A) Response curves against depth, (B) Response curves against temperature, (C) Response curves against distance to shore.

Click here for additional data file.

We thank Global Biodiversity Information Facility (GBIF), Ocean Biogeographic Information System (OBIS), Bio-ORACLE v2.1 dataset and Global Marine Environment Datasets for providing the data to us.

Additional Information and Declarations

Competing Interests

Author Contributions

Data Availability

The authors declare there are no competing interests.

Jinbo Fu conceived and designed the experiments, performed the experiments, analyzed the data, prepared figures and/or tables, authored or reviewed drafts of the paper, and approved the final draft.

Linlin Zhao and Changdong Liu conceived and designed the experiments, authored or reviewed drafts of the paper, and approved the final draft.

Bin Sun analyzed the data, prepared figures and/or tables, and approved the final draft.

The following information was supplied regarding data availability:

The raw georeferenced species data (presence; 1999–2020) are available in the Supplemental Files. The R script we used is also available in the Supplemental Files.

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
