# Peer review of "Estimating the impact of climate change on the potential distribution of Indo-Pacific humpback dolphins with species distribution model"

_PeerJ, doi:10.7717/peerj.12001_

## Round 0.1 · original submission · Minor Revisions

Despite the effort made in data analyses, the reviewers mention drawbacks and limitations, raising some misgivings about the way the manuscript has been written up. They provided constructive comments on how the manuscript can be improved. Furthermore, I included some comments that should be considered. I hope that you will find all advice helpful when revising the manuscript.

(1) English is not always handled well, making some sentences difficult to understand. It needs to be gone over by a fluent speaker to clear up these problems.
(2) The introduction should be better reframed, placing the work in a broad context. Furthermore, the authors need to present a clear hypothesis.
(3) Discussion section (Item 4.4) - Authors need to inform, based on the results, clearer conservation plans considering both short-term and long-term perspectives.

·

Basic reporting

The manuscript has some issues in the clarity, it needs some references and I have some comments about the figures. Please, see "General comments for the author" section for all my comments.

Experimental design

I have some comments about the methods. Please, see "General comments for the author" for more information.

Validity of the findings

No comments.

Additional comments

The research has valuable importance for biodiversity. I am not able to review the language, but the text is, in general, well written, and understandable. I include some suggestion to contribute to the manuscript, please find below:

General suggestion: My suggestion is to change the IPHD for Sousa chinensis or S. chinensis throughout the text.

Abstract:
Clear and objective, it is given the important and relevant information of the manuscript.
Line 26, the information "As an IUCN critically endangered species", is not correct. At the IUCN (and accordingly with the introduction of the present manuscript) the category of Sousa chinensis is VULNERABLE. Please, correct.

Lines 28 and 29: What is the relevance of the term "to our knowledge"? I think that it is implicit, and you do not need to add this term in the abstract. If you remove it, your sentence will be clearer and objective.

Line 28: The threats of human disturbance and environmental pollution to this population have been documented extensively.
In the introduction section (see the lines below) you only give one reference for the extensive documentation, please provide more references or change this sentence.

Key words: The keywords must be more objective and need to reflect the manuscript content and help it be found online. Suggestion: the words "Species distribution models; Ensemble model and Potential distribution" are implicit and/or present in your title (e.g., title: Estimating the impact of climate change on the potential distribution..." and "[...] dolphins with species distribution model distribution model". My suggestion is that you can add some words about your manuscript (e.g., marine fauna, conservation, greenhouse gas emissions, human impact, etc.). Also, in your results the projections of the dolphin also reveal the gain of habitat, thus I think that the word "Habitat contraction" can be changed to "future projections".

Introduction:
Line 47: "Indo-Pacific humpback dolphins (IPHD), Sousa chinensis, belong to the porpoise family of cetaceans are also known as "mermaids'' and "water pandas".
What are known as mermaids and water pandas? All the cetaceans or Sousa chinensis? Please, be clear.

Lines 50, 51, 52, and 53: These areas, which have intensive commercial fisheries, are usually rapidly developing and are easily polluted by industrial production and the lives of local residents; the corresponding consequences of habitat degradation may lead to population declines or even put this species at risk of extinction (Li, 2020).
I do not understand, what happens to the local residents? And the habitat degradation may lead to which population declines? Which species? Please be clear. Also, the reference is needed before the semicolon (i.e., [...] of local residents (REFERENCE?); [...].

Also, in the abstract, you say "The threats of human disturbance environmental pollution to this population have been documented extensively"
Please, provide more references or modify your abstract.

Line 54: "In recent years, the numbers of records on strandings or deaths of IPHD have increased in China" (REFERENCE).
Please give the reference

Line 55: The threats of human disturbance and environmental pollution to this population have been documented extensively ((IUCN, 2019).
Please, update the IUCN reference.
Also, include the correct citation for the inclusion of Sousa chinensis in the IUCN: Jefferson, T.A., Smith, B.D., Braulik, G.T. & Perrin, W. 2017. Sousa chinensis (errata version published in 2018). The IUCN Red List of Threatened Species 2017: e.T82031425A123794774. https://dx.doi.org/10.2305/IUCN.UK.2017-3.RLTS.T82031425A50372332.en. Downloaded on 26 May 2021.

Line 56 and 57: and understanding the species distribution is a prerequisite for plan formulation (REFERENCE).
What do you want to say to understand the distribution? Do you want to know the species distribution? To predict the species distribution for the future? Please, be clear.
Also, give the reference

LInes58 and 59: A species lives in a certain environmental niche space, so environmental changes greatly influence species distribution (REFERENCE).
Please, give the reference.
Also, why do environmental changes have a great influence? Please, be clear.

Lines 58 to 67: Please rewrite this paragraph. The phrases are lost (e.g., The distribution of top marine predators is related to a variety of environmental determinants, such as ocean depth, salinity, distance to the shore, and sea surface temperatures. Global climate change has caused significant changes in marine environmental conditions over the past decades)
Also, Please, give the references.

Lines 71 and 72: Currently, SDMs are applied broadly in the life and environmental science fields (References).
Please, give the references. In what kind of studies, the sdms are applied? Give some examples.

LIne 75 and 76: Accordingly, the use of SDMs in conservation biology and biodiversity assessments is ever increasing (Araújo et al., 2019).
Please, give more examples here.

Line 77: In this study, we developed SDMs and built an ensemble model.
Ensemble for the algorithm or the AOCMs?
Please, specify.

Line 85 and 86: In addition, it provides an essential reference to solve marine conservation planning problems.
This manuscript has, indubitably, great importance, but I do not think that is an essential reference, because the conservation plans can be elaborated without ECMS studies. My suggestion is: "It provides an ADDITIONAL (or important) reference to solve marine conservation planning problems''.

Material and Methods:

Line 89: The figure does not have that information. Please see the lines below.

Line 123: e.g.
E.g., ('exempli gratia') means "for example". You just used three AOGCMs, the correct term is "i.e.," (id est) that means "namely".

Line 124: CSM4, HadGEM2-ES, MIROC5.
Please, provide the institutions where the AOGCMs were generated.

Line 148: To integrate the advantage of each model, we built an ensemble model that was based on the weighted average of the predictions from the selected models and used this ensemble model to predict IPHD distributions under present and future climate conditions.
The ensembles were also generated for the AOGCM, correct? Please, give this information in the methods.

Line 152: by using a threshold that maximized the TSS value
What threshold was used?

Line 158: variable(Zhang et al., 2019.
Please, add a space between "e" and "(" [i.e., variable (Zhang et al., 2019).].

Results:
Line 166: The figures 3 and 4 are supplementary data.

Line 166 and 167: The AUC and TSS values of any individual model were lower than those of the ensemble model.
Do you have any figures to illustrate? That would be great in the supplementary data.

Line 175: The Figure 5 is also supplementary data.

Line 176 and 177: The Figure 6 is also supplementary data.

Discussion

Lines 195 and 96: Utilizing georeferenced presence/pseudoabsence data and the corresponding environmental data, we innovatively developed an ensemble model for IPHD to predict the present and future potential distributions.
Your study has great importance, but the methodology of ensemble models has been used for many years. Even though this study was the first to use that methodology for Sousa chinensis, I am not sure about that word. You can simply write: "we developed an ensemble model" and it's ok and still important.

Line 208, 209, and 210: we recommend using an ensemble model to predict potential species distributions and habitat suitabilities (Araújo and New, 2007; Thuiller et al., 2009; César and Pedro, 2011; Shabani et al., 2016).
You recommend something in that phrase, so why the references? Are those examples of studies that used ensemble models? Please, be clear.

Line 213: for example, the coast of Malaysia, which is also a suitable habitat for white dolphins
These sentences are not necessary. It does not contribute to the sequence of the text.

Line 226: i.e or e.g.?

Line 231: considerable biological and ecological responses
Please, can you give some examples? What is the difference between the East China Sea to other marine habitats?

Line 245 and 246: Because of the data availability, the realized niche of IPHD, such as human activities and dietary structure, were not considered in this study. Stomach content analyses in previous studies have found that humpback dolphins consume a wide variety of pelagic and demersal fishes (Ning et al., 2020).
The methodology used in this paper does not permit the analysis of biotic variables. My suggestion is to remove this phrase.

Line 257: disorders(Xu et al., 2020).
Please add a space between "s" and "(". i.e: disorders (Xu et al., 2020).


Line 264 and 265: Protected areas have been considered to be an effective in situ strategy for conserving biodiversity and ecosystem services (REFERENCE).
Please, give the reference.

Line 267: (2017-2026))
Please, correct. Remove one ")"

Line 268: The
268 adverse effects of climate change on the protected areas of other animals have been elucidated (D’Amen et al., 2011; Zhang et al., 2020b).
What animals?
Also, include an e.g. before the references or, please, provide more references.

Figures:

Figure 1: What would you like to illustrate with this map? If the distribution of Sousa chinensis, please, include the proper information in the caption of what species you are illustrating the distribution. Also, please, include the occurrence points of Sousa Chinensis.

Figures 2, 3, 4, 5, and 6: My suggestion is that figures become supplementary data.

Figure 7: Please correct: Binary outputs to Binary output.
Figure 7A: Please correct: Binary outputs to Binary output.
It is necessary to inform how you make the binarizations, was it based on which thresholds? 10%? Lower presence training?


End of revision

Reviewer 2 ·

Basic reporting

“Estimating the impact of climate change on the potential distribution of Indo-Pacific humpback dolphins with species distribution models” is a good manuscript. The study brings up a worrying scenario for the future conservation of the species, the analyses are largely adequate, and the text is professionally written. However, some adjustments are necessary for different parts of the manuscript for its effective publication. My first comment is the conflict in the number of occurrence records in Line 31 on the Abstract (83 occurrence records) while in Line 96 on Material and Methods (82 of which were within our study area) that should be checked. The Abstract could inform more details about the timelapse to forecast under different future climate scenarios. The Introduction used adequate references. I suggest inserting some references to Line 46-47 and Line 58-59. I think that Objectives (Line 77-86) can be rewritten to make it clearer the real goals and expectations of the manuscript. We already have forecasting models for the target species in a smaller geographic area and with fewer algorithms. Citing them can help in developing hypotheses to answer with a weighted ensemble model, which does not always get the best results as suggested in Line 207 to Line 210. I suggest Vasconcelos & Doro 2016 (10.1007/s11284-016-1401-8) as an example to reformulate de the goals of the study.

Experimental design

The experimental design is very well described for the Study Area, IPHD Data Collection, Environmental Variables, Resolution, and Future Projections. But the description of the weight ensemble model could be improved. Only in the legend of Figures 3 and 4, the reader can find out the threshold for AUC and TSS were select to excluded models that have been utilized in the weight ensemble model, cited in Line 166 in the Results. It was not clear if the ensemble model was made in the biomod 2 package as well. I would like to point out the ODMAP protocol (https://odmap.wsl.ch/ ) that provides a checklist for authors detailing key steps for model building.

Validity of the findings

In Line 37 and in Table 2, the authors mention that over 80% of the suitable habitat in the present day will be lost in all RCP scenarios in the future. This is a worrisome scenario, but the results may be inflated by the high overprediction of the models due to the environmental variables used. Unfortunately, there are no biotic variables for the study area that could refine the model for the target species (see Hunt et al. 2020 cited in the references). Areas, where the species has never been observed in the current climate conditions and were lost in the future climate scenarios, may inflate these results. For conservationist proposals, stability areas are important to report. In Table 2, the authors could report the percentages of new areas and areas of stability to Sousa chinensis, if it is possible. A minor comment is about the known absence points. The authors do not point out whether the known absence points were correctly predicted by the models for the current climate conditions. This could strengthen the validity of the models.

Additional comments

Figure 1 could be more informative, or it could be deleted. The graphs in Figures 3 and 4 are usually displayed as box plots. The article should avoid indicating that ensemble models are the best as quoted in lines 207 and 210. Hao et al. (2020) demonstrate a case where this type of model has not shown the best performance over individual models (https://onlinelibrary.wiley.com/doi/epdf/10.1111/ecog.04890). Ensemble models are important to reduce uncertainties, but it should be emphasized that evaluate the predictive ability of models is just as important.

---

## Round 0.2 · accepted · Accept

The authors have done a commendable job of responding to the comments pointed out, and I’m glad to recommend acceptance of this manuscript. However, consider making the following changes during the proof correction step:
(1) Abstract - avoid using acronyms (e.g., TSS, AUC, RCP) or inform what they mean
(2) Table 1 – inform what the values in parentheses mean